# Analysing trajectories of a longitudinal exposure: A causal perspective on common methods in lifecourse research

**Sarah C. Gadd** [1,2]*, **Peter W. G. Tennant**[1,3,4], **Alison J. Heppenstall**[1,2,4], **Jan R. Boehnke**[5], **Mark S. Gilthorpe**[1,3,4]

1 Leeds Institute of Data Analytics, University of Leeds, Leeds, England, United Kingdom, 2 School of Geography, University of Leeds, Leeds, England, United Kingdom, 3 School of Medicine, University of Leeds, Leeds, England, United Kingdom, 4 The Alan Turing Institute, London, England, United Kingdom, 5 School of Nursing and Health Sciences, University of Dundee, Dundee, Scotland, United Kingdom

* S.C.Gadd@leeds.ac.uk

**Data Availability Statement:** Data were simulated using R code which is available as supplementary material (S1 Appendix).

## Abstract

Longitudinal data is commonly analysed to inform prevention policies for diseases that may develop throughout life. Commonly methods interpret the longitudinal data as a series of discrete measurements or as continuous patterns. Some of the latter methods condition on the outcome, aiming to capture 'average' patterns within outcome groups, while others capture individual-level pattern features before relating these to the outcome. Conditioning on the outcome may prevent meaningful interpretation. Repeated measurements of a longitudinal exposure (weight) and later outcome (glycated haemoglobin levels) were simulated to match three scenarios: one with no causal relationship between growth rate and glycated haemoglobin; two with a positive causal effect of growth rate on glycated haemoglobin. Two methods that condition on the outcome and one that did not were applied to the data in 1000 simulations. The interpretation of the two-step method matched the simulation in all causal scenarios, but that of the methods conditioning on the outcome did not. Methods that condition on the outcome do not accurately represent a causal relationship between a longitudinal pattern and outcome. Researchers considering longitudinal data should carefully determine if they wish to analyse longitudinal data as a series of discrete time points or by extracting pattern features.

## Introduction

Lifecourse data comprise longitudinal data (repeated measurements) that span some or all of life. Analyses of lifecourse data are popular for informing preventative policies to improve population health and wellbeing [1]. For example, temporal patterns of growth (recorded in repeated measures of weight) throughout childhood might be related to risk of type-2 diabetes by age 40 years to target preventative measures at those with certain 'high risk patterns'. To do this effectively, results from analyses must truly reflect a relationship between patterns of growth and diabetes. This may not be the case for some commonly used lifecourse methods.

**Funding:** This work was supported by the Economic and Social Research Council (esrc.ukri. org) [ES/P000746/1 to S.C.G.]; and the Alan Turing Institute (turing.ac.uk) [EP/N510129/1 to P.W.G.T. and M.S.G., ES/R007918/1 to A.H.]. The funders had no role in study design, data collection and analysis, decision to publish, or preparation of the manuscript.

**Competing interests:** The authors have declared that no competing interests exist.

Repeated measurements of a longitudinal exposure, such as weight throughout infancy, are usually correlated with each other, a phenomenon known as autocorrelation [2]. Therefore, they do not satisfy the requirement for independence of observations needed for many common statistical analyses [3]. Methods for capturing and analysing longitudinal exposures typically aim to describe how different patterns of the exposure (e.g. rate of adolescent weight growth) relate to the outcome. Alternatively, some methods aim to identify specific times or 'critical periods' during which the (causal) effect of the exposure is especially strong, or estimate the cumulative effect over multiple exposure times [4, 5].

Generalised methods (g-methods), such as marginal structural models, and methods that explicitly examine "lifecourse hypotheses" offer the most obvious solution to achieving these objectives, given their theoretical foundation within an explicit causal framework [4, 5]. G-methods are however very rarely utilised in applied research, perhaps due to perceived complexity [6]. Simpler and more common methods are less likely to incorporate causal thinking; focussing instead on estimating non-causal associations that are consequently less useful for informing policies and interventions [7, 8].

With the aid of simulations, this paper explains how lack of causal thinking in analyses of longitudinal exposures in relation with later-life outcomes can lead to interpretational biases. Methods that can lead to these biases are compared to an alternative approach that avoids them. This alternative method, however, is not suitable for all situations; other methods, such as g-methods, would be necessary in the presence of time-varying confounding, which is not examined in this paper.

## Methods

Data were simulated to represent the illustrative example of weight measured yearly from birth until age 2 years (the exposure) and diabetes diagnosed at age 40 years (the outcome) from percentage glycated haemoglobin (HbA1c) [9]. This is analogous to routinely-collected health data or data from birth cohorts. Three illustrative scenarios with different causal structures were simulated matching the directed acyclic graphs in Fig 1. Each arrow specifies a direct causal relationship between variables. The absence of an arrow means there is no direct causal relationship, but there may still be a correlation. In Scenario A, birthweight causes HbA1c and there is no causal effect of growth rate on HbA1c. In Scenario B $weight_1$ directly causes HbA1c and growth rate indirectly causes HbA1c through $weight_1$. In Scenario C $weight_2$ directly causes HbA1c and growth rate indirectly causes HbA1c through $weight_2$. For ease of illustration, confounding (by e.g. genetics or *in utero* nutrition) was represented by a single unmeasured common cause of birthweight and growth (U).

1000 datasets comprising 1000 observations were simulated using R 3.4.3; exceeding the number required to achieve >99% accuracy for the parameters of interest [10]. Simulation code is available in the S1 Appendix. Each directed acyclic graph was converted into a covariance matrix of the weight and HbA1c variables using the parameters in Table 1 and standardised path coefficients in Fig 1. Data were simulated with multivariate normal distributions.

Linear growth was simulated for ease of interpretation. HbA1c was dichotomised into a binary variable at the National Institute for Health and Care Excellence threshold for diagnosing type-2 diabetes (HbA1c > 6.5%) [11]. The mean and standard deviation (SD) of simulated weight values, along with the correlation of each weight measure with HbA1c, were averaged across all simulations with 2.5th and 97.5th centiles depicting empirical 95% confidence intervals (CIs).

Data were analysed using three methods: Z-score plots, multilevel models (outcome as covariate), and multilevel models (two-step). Z-score plots are a simple, graphical approach

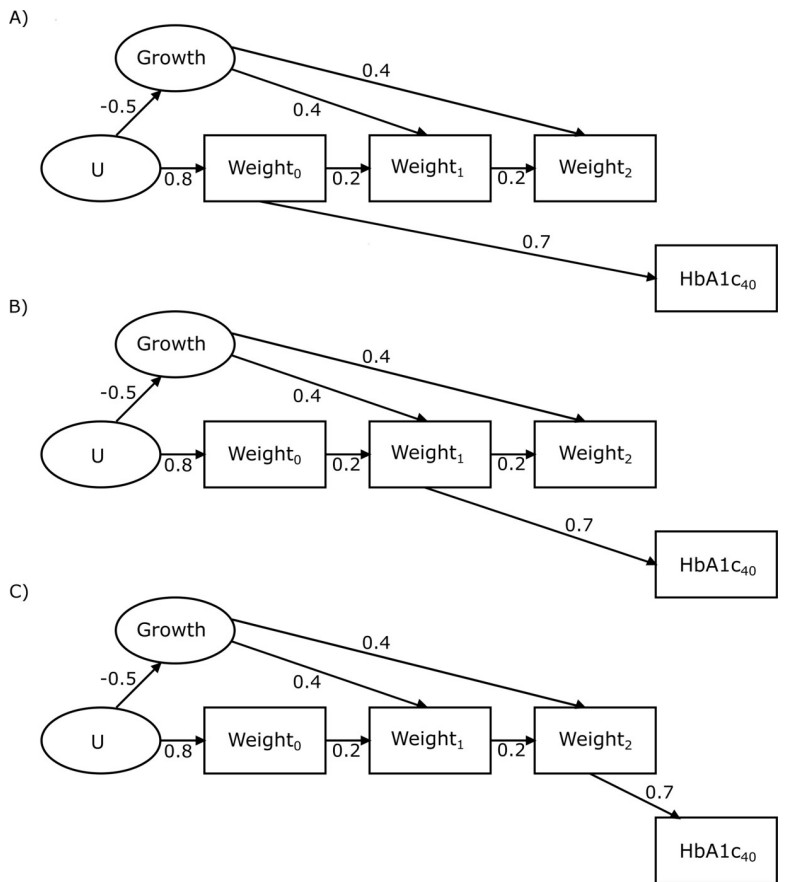

**Fig 1. Directed acyclic graph showing the structure of causal relationships between variables in simulated scenarios A, B and C.** Growth represents the growth rate of an individual and is not simulated or measured in the scenario. U is an unknown and unmeasured variable. The age in years at which known variables are measured is shown in subscript. Arrows show the direction of causal relationships and numbers attached to these arrows show the correlations induced by them.

that aims to identify exposure patterns that lead to an outcome [9, 12]. Weight at each age was standardised into z-scores using the time-specific sample mean and SD [11]. The mean z-scores in those who did and did not develop diabetes were then plotted at each age and connected. Z-score plots are often viewed and interpreted as 'patterns' of weight that 'lead to' the outcome [13]. The plots presented show mean values from all simulations, with $2.5^{th}$ and $97.5^{th}$ centiles depicting empirical 95% CIs.

The multilevel model (outcome as covariate) analysis involved fitting multi-level models of weight over time, with covariates for age, diabetes status, and an age-diabetes interaction term,

**Table 1. Parameters of latent variables and error terms used to simulate data in section 3.**

| Variable | $Weight_0$ (kg) | $Weight_1$ (kg) | $Weight_2$ (kg) | HbA1c (%) |
|---|---|---|---|---|
| Mean | 4 | 8 | 12 | 5.8 |
| Standard deviation | 2 | 2 | 2 | 1 |

The path diagram used to generate observed variables from these is shown in Fig 1.

defined by the following equations (where *i* indexes observations and *j* individuals):

$$weight_{ij} = \beta_{0j} + \beta_1 time_{ij} + \beta_2 diabetes_j + \beta_3 time_{ij} * diabetes_j + \varepsilon_{ij}$$

$$\beta_0 = \gamma_0 + u_{0j}$$

$$\beta_1 = \gamma_1 + u_{1j}$$

$$var\,{}^{u_{0j}}_{u_{1j}} = \begin{pmatrix} \sigma^2_{u0} & \sigma_{u01} \\ \sigma_{u01} & \sigma^2_{u1} \end{pmatrix} \qquad (1)$$

Intercept and age coefficients were free to vary randomly between individuals. Age was centred at one year [14, 15]. A first-order autocorrelated error structure was specified to account for the effect of each weight measure on the subsequent. Multilevel models like this are typically interpreted from the coefficient of the interaction term; for example, a positive interaction between diabetes and time would be interpreted as meaning that an increased growth rate leads to diabetes. Coefficients for these interaction terms were recorded over the 1000 simulations to obtain a median and empirical 95% CIs.

The multilevel model (two-step) approach involved fitting two models, defined by the following equations (where *i* indexes observations and *j* individuals):

$$weight_{ij} = \beta_{0j} + \beta_1 time_{ij} + \varepsilon_{ij}$$

$$\beta_0 = \gamma_0 + u_{0j}$$

$$\beta_1 = \gamma_1 + u_{1j}$$

$$var \begin{pmatrix} u_{0j} \\ u_{1j} \end{pmatrix} = \begin{pmatrix} \sigma^2_{u0} & \sigma_{u01} \\ \sigma_{u01} & \sigma^2_{u1} \end{pmatrix} \qquad (2.1)$$

$$\log\left(\frac{P(diabetes)}{1 - P(diabetes)}\right) = \beta_3 \beta_1 + \varepsilon$$

$$\varepsilon \sim N(0, \sigma^2_\varepsilon) \qquad (2.2)$$

The first (Eq 2.1) was a multilevel model of weight by age, with a first order autocorrelated error structure, to estimate growth as depicted in each directed acyclic graph in Fig 1. The intercept and age coefficients were permitted to vary randomly across individuals and age was centred. The individual-level age coefficients were recorded, representing individuals' growth rates. In the second step (Eq 2.2), a logistic regression model was fitted with diabetes as the outcome, the age coefficient (growth rate) as the exposure, and a birthweight covariate to condition for its confounding influence [16]. The exponentiated model coefficients represent the change in odds of developing diabetes for each increase of 0.1kg/year (selected due to the small growth rate). Coefficients greater than one suggest that higher growth rates lead to diabetes. Coefficient point estimates for the growth rate exposure were recorded to obtain a median and empirical 95% CI from the 2.5[th] and 97.5[th] centiles over the 1000 simulations. All multilevel models were fitted using R package 'nlme' [17].

**Table 2. Summary of simulated variables in scenario A.**

| | $Weight_0$ (kg) | | $Weight_1$ (kg) | | $Weight_2$ (kg) | | $HbA1c_{40}$ (%) | |
|---|---|---|---|---|---|---|---|---|
| | Mean | 95%CI | Mean | 95%CI | Mean | 95%CI | Mean | 95%CI |
| Mean | 4.003 | 3.879, 4.129 | 8.003 | 7.889, 8.133 | 11.997 | 11.863, 12.117 | 5.800 | 5.737, 5.862 |
| SD | 2.000 | 1.910, 2.093 | 1.998 | 1.910, 2.090 | 2.000 | 1.911, 2.092 | 0.999 | 0.955, 1.043 |
| Correlation with HbA1c | 0.699 | 0.664, 0.729 | 0.029 | -0.033, 0.091 | -0.105 | -0.167, -0.044 | 1 | |

95%CI represents 95% empirical confidence intervals.

Any errors from the multilevel model (outcome as covariate) and multilevel model (two-step), such as failure to converge, were recorded, and the estimates from these datasets were discarded.

## Results

One dataset for each of scenarios A and B, and 20 datasets in scenario C were discarded due to models failing to converge. The mean and SD of weight at each time, averaged across all remaining simulations for each scenario are shown in Tables 2, 3 and 4, along with mean correlations of each weight measure with HbA1c. In scenario A, there was a large positive correlation at birth, decreasing to a small positive correlation at age 1, and a small negative correlation at age 2. In scenario B, there was a near-zero correlation at birth, increasing to a large positive correlation at age 1, and decreasing to a small positive correlation at age 2. In scenario C, there is a small negative correlation at birth, increasing to a small positive correlation at age 1, and a large positive correlation at age 2.

The z-score plots for each scenario are shown in Fig 2. In scenario A, the diabetic group has a much higher weight at birth and the points on the graph are far apart, and far from the overall mean (zero). The points converge over time until they meet, cross and begin to diverge between age 1 and 2 years. By age 2, the diabetic group have a lower mean weight z-score than the non-diabetic group. In scenario B, the points are close to the overall mean at birth, diverging substantially at age 1, before converging back towards the mean at age 2; the diabetic group always has a higher mean weight z-score than the non-diabetic group. In scenario C, the diabetic group starts with a slightly lower birthweight than the non-diabetic group, but the z-score increases over time, while the non-diabetic group decreases, leading to a large difference at age 2.

Results from the multilevel models (outcome as covariate) are in Table 5 and Fig 3, which show the model-fitted regression lines and true mean weight values for the diabetic and non-diabetic groups at each time point. The model values do not always fit well with the mean values (see especially scenario B in Fig 3B) because the models were constrained to linearity (because *growth* was simulated to be linear for simplicity), but the mean values in each

**Table 3. Summary of simulated variables in scenario B.**

| | $Weight_0$ (kg) | | $Weight_1$ (kg) | | $Weight_2$ (kg) | | $HbA1c_{40}$ (%) | |
|---|---|---|---|---|---|---|---|---|
| | Mean | 95%CI | Mean | 95%CI | Mean | 95%CI | Mean | 95%CI |
| Mean | 4.001 | 3.870, 4.124 | 7.999 | 7.876, 8.127 | 11.997 | 11.885, 12.118 | 5.801 | 5.741, 5.859 |
| SD | 2.000 | 1.912, 2.088 | 2.000 | 1.917, 2.084 | 1.999 | 1.915, 2.09 | 1.000 | 0.956, 1.044 |
| Correlation with HbA1c | 0.027 | -0.034, 0.088 | 0.699 | 0.666, 0.731 | 0.229 | 0.169, 0.283 | 1 | |

95%CI represents 95% empirical confidence intervals.

**Table 4. Summary of simulated variables in scenario C.**

| | Weight$_0$ (kg) | | Weight$_1$ (kg) | | Weight$_2$ (kg) | | HbA1c$_{40}$ (%) | |
| --- | --- | --- | --- | --- | --- | --- | --- | --- |
| | Mean | 95%CI | Mean | 95%CI | Mean | 95%CI | Mean | 95%CI |
| Mean | 3.997 | 3.873, 4.113 | 8.003 | 7.882, 8.122 | 12.001 | 11.875, 12.122 | 5.801 | 5.738, 5.861 |
| SD | 2.001 | 1.919, 2.087 | 2.000 | 1.911, 2.087 | 2.003 | 1.915, 2.101 | 1.001 | 0.959, 1.047 |
| Correlation with HbA1c | -0.106 | -0.166, -0.043 | 0.229 | 0.167, 0.288 | 0.700 | 0.666, 0.731 | 1 | |

95%CI represents 95% empirical confidence intervals.

outcome group change nonlinearly. In all scenarios, the coefficient of age is positive, confirming that weight increases from birth to age 2. For scenario A, the *negative* age-diabetes interaction term and shallower slope of increasing weight in the diabetes group (Fig 3A) suggests that those who developed diabetes grew slightly *slower* than those who did not develop diabetes. In

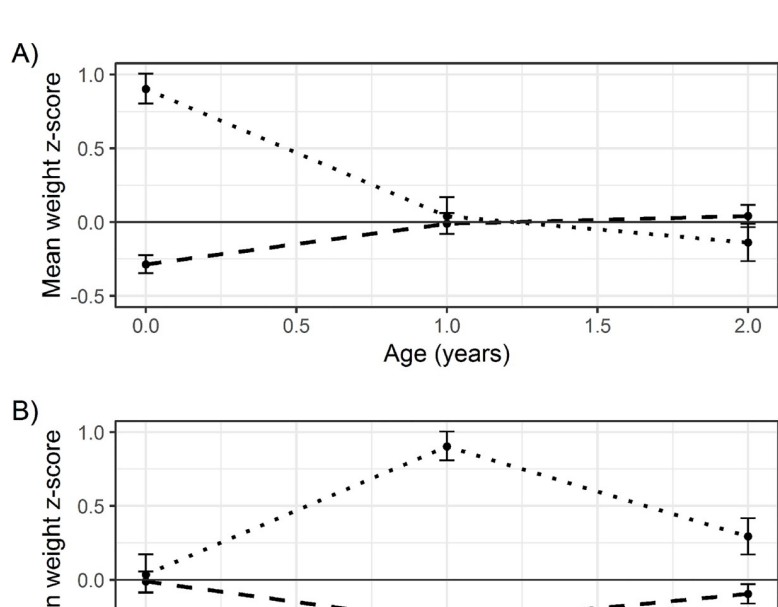

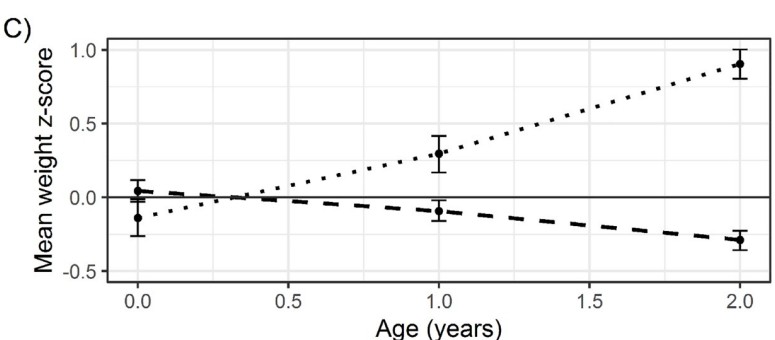

**Fig 2. Z-score plots of weight from birth to age 2 years for scenarios A, B and C.** Dotted lines show the group diagnosed with diabetes at age 40 and dashed those without a diagnosis. Error bars show empirical 95% confidence intervals.

**Table 5. Average parameter estimates from multilevel models of weight (outcome as covariate).**

| Parameter | Scenario A | | Scenario B | | Scenario C | |
|---|---|---|---|---|---|---|
| | Mean | 95% CI | Mean | 95% CI | Mean | 95% CI |
| Diabetes | 0.729 | 0.566, 0.900 | 1.070 | 0.899, 1.25 | 0.939 | 0.769, 1.111 |
| Age | 4.327 | 4.223, 4.430 | 3.914 | 3.807, 4.021 | 3.670 | 3.563, 3.767 |
| Diabetes*Age | -1.372 | -1.572, -1.166 | 0.340 | 0.133, 0.573 | 1.375 | 1.166, 1.573 |
| Intercept | 7.823 | 7.737, 7.911 | 7.740 | 7.655, 7.821 | 7.770 | 7.690, 7.862 |
| Intercept variance | 0.481 | 0.348, 0.601 | 0.503 | 0.127, 0.816 | 0.099 | 0.000, 0.255 |
| Age Variance | 0.680 | 0.547, 0.799 | 0.897 | 0.714, 1.08 | 0.567 | 0.000, 0.711 |
| Residual Variance | 1.770 | 1.710, 1.836 | 1.727 | 1.54, 1.849 | 1.840 | 1.778, 1.908 |
| Constant-Age Covariance | 0.973 | 0.894, 0.988 | 0.268 | 0.02, 0.845 | -0.729 | -0.957, 0.579 |
| Autocorrelation parameter | 0.120 | 0.069, 0.169 | 0.042 | -0.118, 0.134 | 0.160 | 0.119, 0.196 |

scenario B, the small *positive* age-diabetes interaction term and slightly steeper slope (Fig 3B) suggests that those who developed diabetes grew slightly *faster* than those who did not develop diabetes. In scenario C, the large *positive* diabetes-age interaction term and steeper slope (Fig 3C) suggests that those who developed diabetes grew substantially faster than those who did not develop diabetes.

Results from the multilevel models (two-step) are shown in Table 6. In scenario A, the odds ratio for growth rate was 1.000 (95% empirical CI: 0.943, 1.057), suggesting that the odds of diabetes were unaffected by growth rate. In scenario B, the odds ratio was 1.194 (95% empirical CI: 1.122, 1.316), suggesting that the odds of diabetes increased modestly with increasing growth rate. In scenario C, the odds ratio was 1.679 (95% empirical CI: 1.477, 2.191), suggesting the odds of diabetes increased substantially with increasing growth rate.

## Discussion

In Scenario A, we simulated no causal effect of growth rate on the risk of developing diabetes; Birthweight causes HbA1c, but any pattern of growth thereafter is irrelevant. Neither the z-score plot nor the multilevel model (with the outcome as a covariate) reflect this and would be erroneously 'interpreted' as showing that slower growth leads to diabetes. Conversely, the coefficient from the two-step multilevel model correctly implies no effect of growth rate on diabetes risk.

In scenario B, we simulated that weight$_1$ caused diabetes, which could be interpreted as growth causing HbA1c through weight$_1$. The z-score plot however suggests that faster growth up to age 1 *and* slower growth thereafter leads to diabetes. This does not reflect the causal relationship simulated, where higher growth only increased the risk of diabetes by increasing weight at age 1. Both the multilevel model (with the outcome as a covariate) and the two-step multilevel model reflect this more closely, suggesting that higher growth rates caused diabetes.

In scenario C, we simulated that weight$_2$ caused diabetes, which could again be interpreted as growth causing HbA1c through weight$_2$ (and indirectly through weight$_1$). Here, the results from all three methods correctly suggest that higher growth rate cause diabetes.

The z-score plots (and common interpretation thereof) only reflected the simulated truth in one of the three scenarios, revealing this is not a reliable approach for examining the causal effect of a longitudinal exposure on a distal outcome. This is because average weight z-scores at each time point are explicitly calculated and presented within groups of the outcome. By inappropriately conditioning on the outcome in an attempt to examine 'average patterns' of weight associated with diabetes, the method actually examines cross-sectional associations between weight and the outcome at each time point. This problem remains even if only one

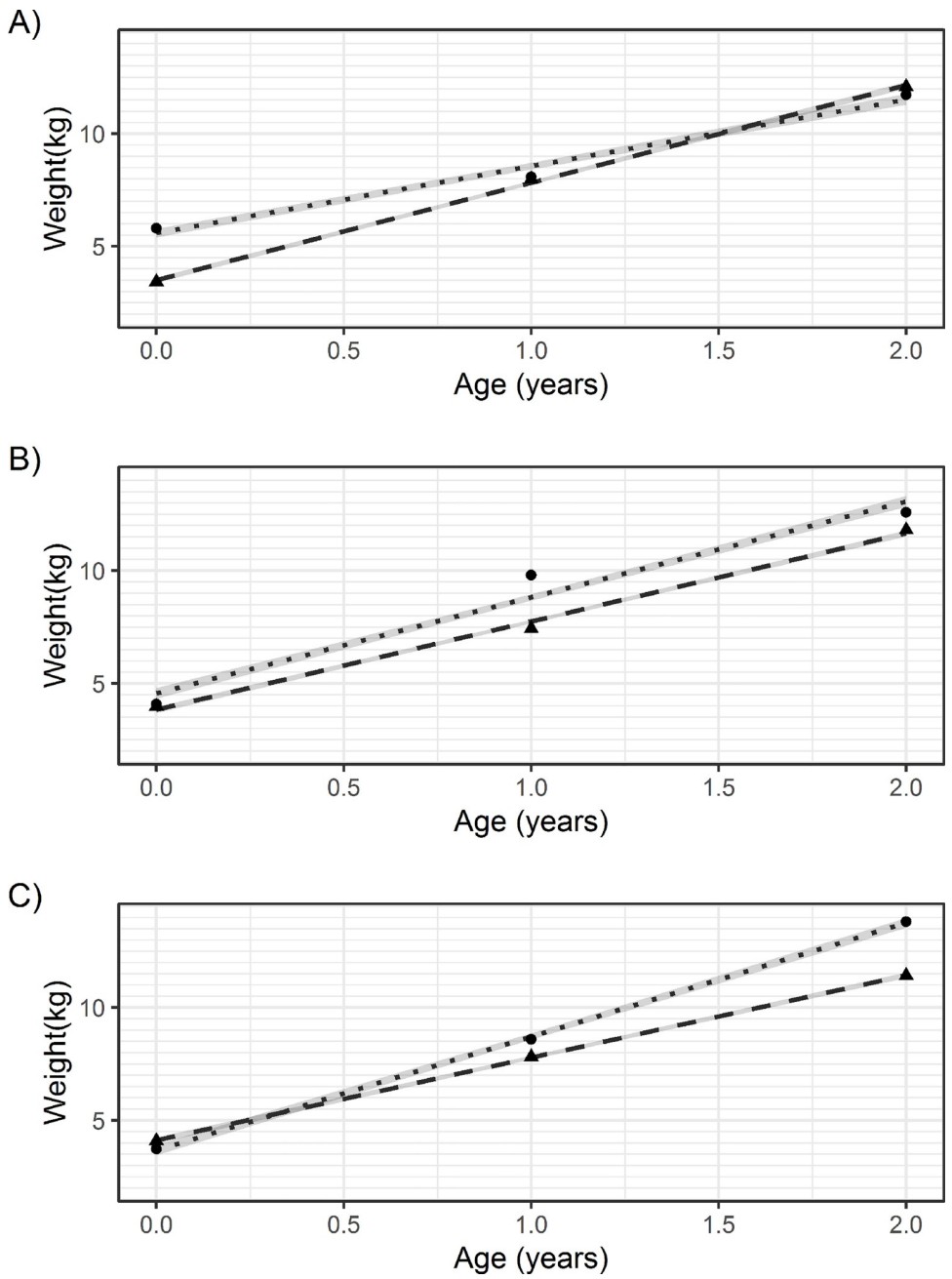

**Fig 3. Fitted weight values from multilevel models (outcome as covariate) and average mean weight values for scenarios A, B and C.** Dotted lines (fitted values) and circular points (average mean weight values) represent fitted values for the group with a diabetes diagnosis at age 40. Dashed lines (fitted values) and triangular points (average mean weight values) represent those without a diagnosis. The grey ribbon represents an empirical 95% confidence band around the fitted values.

group (e.g. those with diabetes) is considered. The value of each mean z-score (e.g. weight) has no obvious causal meaning; instead, it reflects the size of the cross-sectional correlation between the exposure and the outcome at each time point. Because the standardisation process fixes the scale, time points with the strongest cross-sectional correlations will always appear most different, and those with the weakest correlations will always appear most similar. For

**Table 6. Average parameter estimates from the logistic regression model of diabetes status on weight growth rate.**

| Parameter | Scenario A | | Scenario B | | Scenario C | |
|---|---|---|---|---|---|---|
| | Mean | 95% CI | Mean | 95% CI | Mean | 95% CI |
| Growth rate | 1.000 | 0.943, 1.057 | 1.194 | 1.122, 1.316 | 1.679 | 1.477, 2.191 |
| Weight$_0$ | 2.000 | 2.060, 2.745 | 1.000 | 1.149, 1.429 | 2.000 | 1.339, 2.265 |
| Constant | $6.030 \times 10^{-03}$ | $3.654 \times 10^{-04}, 9.185 \times 10^{-02}$ | $9.309 \times 10^{-05}$ | $1.659 \times 10^{-06}, 1.291 \times 10^{-03}$ | $2.221 \times 10^{-11}$ | $2.552 \times 10^{-16}, 7.591 \times 10^{-09}$ |

Growth rate was estimated using a multilevel model of weight over age (agnostic to the outcome, diabetes status)

example, in Scenario A, there was a strong positive correlation between birthweight and diabetes due to the causal effect of birthweight, and weaker correlations at ages 1 and 2 years as the contribution of birthweight to weight decreased. This is reflected by the z-score plot in Fig 2; the mean weight z-scores values are farthest apart at birth and coverage over time. In scenarios B and C, the strongest correlations are at ages 1 and 2 years respectively; the corresponding z-scores plots are likewise farthest apart at these time points. The absolute value of each time point z-score should not therefore be joined or compared to the z-score values at other time points because the 'patterns' that appear have no causal meaning and do not represent individual growth *trajectories*.

Inappropriate conditioning on the outcome also affects multilevel models where the outcome is included as a model covariate. The consequences are not identical to the z-score plot because the scale has not been fixed by standardisation and the correlation pattern is assumed to follow a specific parametric shape. In our example, the linearity constraint introduced misfit between the modelled regression lines and the mean weight values (Fig 3). In Scenario B this meant the model failed to highlight that the largest cross-sectional correlation between weight and diabetes occurred at age 1 year, and this explains the difference in interpretation with the corresponding z-score plot. In scenarios A and C, models were similar enough to the average weight values to provide similar interpretations. Had we simulated nonlinear growth, however, the linearity constraint would likely have introduced further differences in interpretation compared with the z-score plot.

The multilevel model (two-step) approach is more robust than the other approaches because it does not involve conditioning on the outcome. Instead, exposure patterns are modelled and only in the second step are these related to the outcome. This approach genuinely treats the exposure as a longitudinal variable and should therefore be strongly favoured over approaches that condition on the outcome whenever there is an interest in the causal interpretation of a *longitudinal* exposure pattern. This method is not, however, without limitations.

First, because the second step of the multilevel model (two step) approach treats the unobserved growth rate estimates as fully observed, it underestimates the standard errors (and confidence intervals), even when attempts are made to address this [18]. Alternative latent variable methods, like latent growth curve models, growth mixture models, and autoregressive latent trajectory models, which retain the latent, or unobserved, nature of the pattern features avoid this problem.

Second, two-step multilevel models and their constructed latent variable alternatives can still present some interpretational challenges from a causal inference perspective. By summarising the effect of multiple measurements that span a period into one or more average feature (s), such as growth rate, the causal contributions of each individual measurement occasion is lost, as too are any corresponding 'critical' period effects [19]. This places such methods in contrast to G-methods, where the focus is explicitly on estimating the causal effect of the exposure as measured at each time point. Whether the underlying feature (e.g. growth) or the

individual measure (e.g. weight at age 1) are the 'true' cause cannot be distinguished statistically because they are simply different ways of describing the same information. In scenarios B and C, for instance, both 'growth rate' and the individual measures of weight at ages 1 and 2 years respectively appear to cause diabetes. Changing either would therefore change the risk of diabetes, and neither can be described as more or less responsible. The choice of whether to analyse a longitudinal exposure as a series of discrete measures or a summary feature (e.g. growth rate) may therefore be down to philosophical and/or contextual preferences regarding the question(s) posed. That said, since many pattern features like 'growth rate' span several measurement intervals, they are susceptible to time-varying confounding by any variables that are simultaneously caused by earlier measures while causing later measures, i.e. so-called intermediate confounders. In such situations, there may be no alternative to g-methods, which are currently unique for their compatibility with intermediate confounding [6].

It is important to note that, for illustrative purposes, this paper presents a simplified scenario in which there are no competing events or loss to follow up, both of which would be present in reality. Any differential loss to follow up or occurrence of competing events would (further) bias the results from all three methods examined in this paper [20].

## Recommendations

Methods that condition on the outcome are not appropriate for examining the causal relationship between patterns of a longitudinal exposure and a later outcome, as they only describe the cross-sectional correlations at each time point. The apparent 'patterns' that are observed have no causal interpretation and should not be interpreted as individual exposure trajectories that cause the outcome.

Alternative analytical strategies should seek to describe features of the exposure agnostic to the outcome, whether explicitly in two separate models or implicitly using latent variable methods. Researchers should however carefully consider whether pattern features or discrete measures are more appropriate, useful and/or interpretable ways to capture a specific 'exposure' in a specific context. If interested in the effect of exposures at specific 'critical' points in time then alternative methods are recommended [4, 5].

If a pattern feature is truly of interest, researchers should think very carefully about which pattern feature(s) are of interest before analysis. In the absence of a single, distinct and clearly identifiable causal feature it is tempting to consider summarising the 'average' of exposure 'trajectories' for individuals with different outcomes by conditioning on the outcome, but this risks highly misleading results. A longitudinal exposure—or pattern thereof—that spans a long period may be conflated with intermediate confounding and thus fail to describe the true causal process of interest. Features that occur at specific time periods that have a tangible real-world meaning may be best suited to the methods recommended, such as two-step multilevel models.

## Conclusion

This paper explains how longitudinal data analyses that inappropriately condition on the outcome may lead to biased inferences about how exposure patterns affect later outcomes. Methods such as z-score plots and multilevel models with the outcome as a covariate do not create causally meaningful exposure 'patterns' and, as our simulations show, can be highly misleading.

In lifecourse research, or whenever interested in the causal relationship between a longitudinal exposure and later outcome, we recommend avoiding methods that inappropriately condition on the outcome in favour of methods that capture patterns *a priori*, although the potential influence of intermediate confounding should be carefully considered.

## Supporting information

**S1 Appendix. Code used to simulate and analyse data.**
(DOCX)

## Author Contributions

**Conceptualization:** Mark S. Gilthorpe.

**Formal analysis:** Sarah C. Gadd.

**Investigation:** Sarah C. Gadd.

**Methodology:** Sarah C. Gadd, Peter W. G. Tennant, Mark S. Gilthorpe.

**Supervision:** Peter W. G. Tennant, Alison J. Heppenstall, Mark S. Gilthorpe.

**Writing – original draft:** Sarah C. Gadd.

**Writing – review & editing:** Peter W. G. Tennant, Alison J. Heppenstall, Jan R. Boehnke, Mark S. Gilthorpe.

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
