## [Decision Letter · Decision Letter 0]

16 Oct 2019

PONE-D-19-21843

Analysing trajectories of a longitudinal exposure: a causal perspective on common methods in lifecourse research

PLOS ONE

Dear Ms Gadd,

Thank you for submitting your manuscript to PLOS ONE. After careful consideration, we feel that it has merit but does not fully meet PLOS ONE’s publication criteria as it currently stands. Therefore, we invite you to submit a revised version of the manuscript that addresses the points raised during the review process.

We would appreciate receiving your revised manuscript by Nov 30 2019 11:59PM. To enhance the reproducibility of your results, we recommend that if applicable you deposit your laboratory protocols in protocols.io, where a protocol can be assigned its own identifier (DOI) such that it can be cited independently in the future. For instructions see: http://journals.plos.org/plosone/s/submission-guidelines#loc-laboratory-protocols

We look forward to receiving your revised manuscript.

Kind regards,

Prof Stephen Z Levine, PhD

Academic Editor

PLOS ONE

Journal Requirements:

2. In your Methods section, it is reported that "Simulation code is available as supplementary material"; please correct this statement, as we note that the code is not provided as a supplementary file, but is instead deposited in GitHub.

Additional Editor Comments (if provided):

Reviewers' comments:

Reviewer's Responses to Questions

**Comments to the Author**

1. Is the manuscript technically sound, and do the data support the conclusions?

Reviewer #1: Yes

Reviewer #2: Yes

2. Has the statistical analysis been performed appropriately and rigorously? 

Reviewer #1: Yes

Reviewer #2: Yes

3. Have the authors made all data underlying the findings in their manuscript fully available?

Reviewer #1: No

Reviewer #2: Yes

4. Is the manuscript presented in an intelligible fashion and written in standard English?

Reviewer #1: Yes

Reviewer #2: Yes

5. Review Comments to the Author

Reviewer #1: This is an interesting paper, addressing a very important methodological issue in lifecourse epidemiology. I have some suggestions to improve the accessibility of this manuscript:

1. To make the comparison across different models/methodologies more explicit, I think it would be useful to spell out the model specifications in equations. Although the directed acyclic diagrams are useful for visualizing the causal relations between variables, equations help readers with the understanding of the statistical models used to estimate the causal relations.

2. Since DAGs are used to show the causal structure in the simulated data, I think it would be equally useful to use DAGs to show the implied causal structures within the statistical models (e.g. multilevel models etc). SO readers could compare the DAGs to apprehend why some models are more "correct" than others in different scenarios.

3. Although the causal relations between the variables in the simulated data were shown in DAGs, I feel it would useful to provide the R codes in the appendix; so readers could examine the details of the simulations.

Reviewer #2: This is a clear, nicely written paper on an important topic. My comments are below:

Major comment:

1. The methods that condition on the outcome didn't perform as badly as I expected. My understanding is that this is because of the restriction to settings with no time-varying confounder-treatment feedback as stated in the discussion. I think it would be helpful to introduce this restriction / assumption earlier in the paper -- perhaps in the introduction -- so that the results don't get over-generalized.

Minor comment:

1. The simulated data creates a scenario in which no one is lost to follow-up or experiences a competing event. While this is a reasonable simplification to demonstrate the problems with the outcome-control methods, some readers may miss that there are further potential dangers associated with LTFU or competing events. A short review of these issues in the discussion would make the paper a bit more complete.

6. PLOS authors have the option to publish the peer review history of their article (what does this mean?). If published, this will include your full peer review and any attached files.

Reviewer #1: No

Reviewer #2: No

---

## [Author Response · Author response to Decision Letter 0]

28 Oct 2019

Reviewer #1 Questions

1. Is the manuscript technically sound, and do the data support the conclusions?

Reviewer #1: Yes

2. Has the statistical analysis been performed appropriately and rigorously?

Reviewer #1: Yes

3. Have the authors made all data underlying the findings in their manuscript fully available?

Reviewer #1: No

Author response: The data underlying findings in this manuscript was simulated using code made available on GitHub which we appreciate was not correctly indicated in the manuscript. Following on from this, we decided to include code as an appendix as GitHub hosting may not be permanent.

Author change: Changed “as supplementary material.” to “in the S1 appendix.” in Methods [p5]; Added “Supporting Information Captions¶ S1 Appendix: Code used to simulate and analyse data.” at end of manuscript [p19]. Supplementary material has been submitted in the file “S1_Appendix.docx”.

4. Is the manuscript presented in an intelligible fashion and written in standard English?

Reviewer #1: Yes

Reviewer #1 Comments to the Author

This is an interesting paper, addressing a very important methodological issue in lifecourse epidemiology. I have some suggestions to improve the accessibility of this manuscript:

1. To make the comparison across different models/methodologies more explicit, I think it would be useful to spell out the model specifications in equations. Although the directed acyclic diagrams are useful for visualizing the causal relations between variables, equations help readers with the understanding of the statistical models used to estimate the causal relations.

Author response: Thank you for this suggestion, we agree that this will be a useful addition to the paper to clarify the models used. We note that no equations have been provided for the Z-score plots as these do not use models.

Author change: Equation 1 added following the changed sentence: “The multilevel model (outcome as covariate) analysis involved fitting multi-level models of weight over time, with covariates for age, diabetes status, and an age-diabetes interaction term, defined by the following equations (where i indexes observations and j individuals):“ in Methods [p6]; Equation 2 added following the altered sentence: “The multilevel model (two-step) approach involved fitting two models, defined by the following equations (where i indexes observations and j individuals):” in Methods [p6-7]. References to Equation 2.1 and 2.2 added in Methods [p7]: “The first (Equation 2.1) was a multilevel…” and “In the second step (Equation 2.2), a logistic…”

2. Since DAGs are used to show the causal structure in the simulated data, I think it would be equally useful to use DAGs to show the implied causal structures within the statistical models (e.g. multilevel models etc). SO readers could compare the DAGs to apprehend why some models are more "correct" than others in different scenarios.

Author response: Thanks for this useful suggestion. While we agree that visual representations of models are often useful for explaining them, in this case, we have chosen not to include DAGs to represent the implied causal structures of the models for the following reasons: 

1. A single model specification could correctly estimate a causal effect for a multitude of different DAGs. That is, a DAG (given a chosen exposure and outcome) implies one (or several) appropriate models, but a given model may also be appropriate for many DAGs. For example, the four DAGs shown in the Response to Reviewers document all imply the model D~A to find the total causal effect of A on D, but the model implies any of these DAGs (and many more with these four variables). 

We thought to include many implied DAGs for each model could potentially lead to greater confusion than clarity and would certainly warrant a more detailed explanation that might only detract from the main message of the paper. 

2. Some features of the models are not commonly understood as a feature of DAGs. In particular, this relates to the multilevel model with an outcome as covariate which contains an interaction term, indicating “effect modification” between the time and diabetes variables. While effect modification in DAGs has been discussed (https://www.ncbi.nlm.nih.gov/pubmed/17700242) it is not commonly implemented and so would not be a familiar component of a DAG to most readers. 

In the presence of the newly included model equations to clarify their specification (as per comment 1) we felt that the inclusion of implied DAGs may create more confusion than the clarity it would seek to bring.

3. Although the causal relations between the variables in the simulated data were shown in DAGs, I feel it would useful to provide the R codes in the appendix; so readers could examine the details of the simulations.

Author response: Thank you for highlighting this issue – we have made appropriate changes to include the code as supplementary material as discussed in our response to Reviewer #1 Questions #3.

Reviewer #2 Questions

1. Is the manuscript technically sound, and do the data support the conclusions?

Reviewer #2: Yes

2. Has the statistical analysis been performed appropriately and rigorously?

Reviewer #2: Yes

3. Have the authors made all data underlying the findings in their manuscript fully available?

Reviewer #2: Yes

4. Is the manuscript presented in an intelligible fashion and written in standard English?

Reviewer #2: Yes

Reviewer #2 Comments to the Author

This is a clear, nicely written paper on an important topic. My comments are below:

Major comment:

1. The methods that condition on the outcome didn't perform as badly as I expected. My understanding is that this is because of the restriction to settings with no time-varying confounder-treatment feedback as stated in the discussion. I think it would be helpful to introduce this restriction / assumption earlier in the paper -- perhaps in the introduction -- so that the results don't get over-generalized.

Author response: Thank you for this useful comment, we agree that it is important to introduce this earlier in the paper for the reasons stated and we have made changes to do so.

Author changes: In the Introduction section [p4] the sentence “Approaches are suggested to avoid these biases” has been changed to “Such methods are compared to an alternative approach that avoids these biases. This method, however, is not suitable for all situations; other methods, such as g-methods, would be necessary in the presence of time-varying confounding, which is not examined in this paper.”

Minor comment:

1. The simulated data creates a scenario in which no one is lost to follow-up or experiences a competing event. While this is a reasonable simplification to demonstrate the problems with the outcome-control methods, some readers may miss that there are further potential dangers associated with LTFU or competing events. A short review of these issues in the discussion would make the paper a bit more complete.

Author response: This is another important restriction that should be highlighted and we are grateful for pointing it out. 

Author changes: We have added the following paragraph to the discussion section [p15]: “It is important to note that, for illustrative purposes, this paper presents a simplified scenario in which there are no competing events or loss to follow up, both of which would be present in reality. Any differential loss to follow up or occurrence of competing events would (further) bias the results from all three methods examined in this paper [20].” Reference number 20 has been added to the reference list [p19].

---

## [Editor Report · Decision Letter 1]

31 Oct 2019

Analysing trajectories of a longitudinal exposure: a causal perspective on common methods in lifecourse research

PONE-D-19-21843R1

Dear Dr. Gadd,

We are pleased to inform you that your manuscript has been judged scientifically suitable for publication and will be formally accepted for publication once it complies with all outstanding technical requirements.

With kind regards,

Stephen Z Levine, PhD

Academic Editor

PLOS ONE
---

## [Editor Report · Acceptance letter]

7 Nov 2019

PONE-D-19-21843R1 

Analysing trajectories of a longitudinal exposure: a causal perspective on common methods in lifecourse research 

Dear Dr. Gadd:

I am pleased to inform you that your manuscript has been deemed suitable for publication in PLOS ONE. Congratulations! Your manuscript is now with our production department. 

With kind regards,

on behalf of

Professor Stephen Z Levine 

Academic Editor

PLOS ONE